# Unplanned Out-of-Hospital Birth—Short and Long-Term Consequences for the Offspring

**DOI:** 10.3390/jcm9020339

**Published:** 2020-01-25

**Authors:** Gil GUTVIRTZ, Tamar WAINSTOCK, Daniella LANDAU, Eyal SHEINER

**Affiliations:** 1Department of Obstetrics and Gynecology, Soroka University Medical Center, Ben-Gurion University of the Negev, Beer-Sheva POB 151, Israel; sheiner@bgu.ac.il; 2Department of Public Health, Faculty of Health Sciences, Ben-Gurion University of the Negev, Beer-Sheva POB 653, Israel; wainstoc@bgu.ac.il; 3Department of Neonatology, Soroka University Medical Center, Ben-Gurion University of the Negev, Beer-Sheva POB 151, Israel; DanielleLa@clalit.org.il

**Keywords:** out-of-hospital, unplanned, accidental, mortality, hospitalization, follow-up, long-term

## Abstract

The unpredictable nature of childbirth infrequently results in unplanned out-of-hospital birth, in a pre-hospital setting. We evaluated the perinatal and long-term outcome of children accidentally born out-of-hospital. This was a population-based analysis of singleton deliveries occurring at a single tertiary hospital. The maternal characteristics and pregnancy outcome of unplanned out-of-hospital births were compared with in-hospital attended deliveries. Long-term cumulative incidence of hospitalizations (up to 18 years) involving respiratory, neurological, endocrine or infectious morbidity were evaluated using Kaplan–Meier survival curves and Cox regression models were used to control for confounders. In total, 243,682 deliveries were included, and 1.5% (*n* = 3580) were unplanned out-of-hospital births. Most occurred in multiparous women, and about a quarter of these women had inadequate prenatal care. Perinatal mortality rate was significantly higher for out-of-hospital births as compared with in-hospital births (OR = 2.9; 95% CI 2.2–3.8, *p* < 0.001). Kaplan–Meier survival curves demonstrated a significantly lower cumulative incidence of hospitalizations of children born out-of-hospital and the Cox models showed that hospitalization rates involving any of the above morbidities were significantly lower in children born out-of-hospital. While perinatal mortality was higher in unplanned out-of-hospital births, offspring born out-of-hospital showed a lower incidence of hospitalizations involving a variety of morbidities, possibly owing to under-utilization of healthcare services in this population.

## 1. Introduction

Unplanned or accidental out-of-hospital births are, by nature, an undesired event [1]. While the safety of planned out-of-hospital birth is debatable in the literature [2,3], outcomes for unplanned out-of-hospital are worrisome [1]. Perinatal mortality rates for unplanned out-of-hospital birth are two–tjree fold higher than in-hospital birth [1,4]. 

The prevalence of out-of-hospital birth varies, mainly depending on geographical status. Although the out-of-hospital delivery rate may be as low as 0.15% in urban areas, it can reach up to 3% in more rural areas [5]. Accordingly, many out-of-hospital deliveries occur in rural regions inaccessible to medical centers, or areas of rural poverty, where prenatal care is usually underutilized and inadequate [1]. Indeed, risk factors for accidental out-of-hospital are multiparity, inadequate prenatal care, unemployment and long travel time from home to delivery unit [6,7]. Importantly, a significant proportion of unplanned out-of-hospital deliveries occur preterm, which is also probably a major contributor to the higher mortality of these infants. It is believed that if these deliveries were to occur in-hospital, some perinatal mortality cases may have been prevented [8]. 

The short-term perinatal outcomes associated with unplanned out-of-hospital births have been previously investigated, suggesting that surviving babies born out-of-hospital are at higher risk for NICU admission, due to hypothermia, polycythemia, hypoglycemia and convulsions [9]. Whether these adverse outcomes affect the long-term morbidity of the offspring has not yet been determined. In this large population-based study, we followed children until the age of 18 to evaluate the long-term consequences of being born accidentally out-of-hospital.

## 2. Methods

This is a population-based retrospective study of singleton deliveries registered in a single medical center between 1991 and 2014. The study was conducted at the Soroka University Medical Center (SUMC), the sole tertiary medical center in the Negev (southern Israel) and the largest birth center in the country. The Negev region occupies approximately 60% of Israel’s total landmass and SUMC serves the entire population of the region, which is very heterogeneous and consists of 2 major populations—the Jewish and Bedouin populations. This study compared unplanned out-of-hospital deliveries with in-hospital deliveries. Of note, virtually all newborns are brought to the hospital even if delivered out of the hospital, mainly because registration of the baby within 24 h of delivery entitles the mother to birth payment from the government [1]. Post-delivery hospitalization for the mother and the child is also universally funded by the government. This study was approved by the institutional review board (SUMC IRB Committee). 

Data were collected from two separate databases (obstetrical and pediatric) that were cross-linked and merged based on patients’ ID number. The perinatal database consists of information recorded by a physician at admission and immediately following delivery. The pediatric database consists of demographic information and ICD-9 codes for all medical diagnoses made during hospitalizations in SUMC pediatric departments. Medical secretaries routinely review the information and assess medical prenatal care records as well as routine hospital documents, prior to entering it into the database, to insure its maximal completeness and accuracy. 

The primary exposure was unplanned out-of-hospital delivery. Rates of preterm delivery (PTD—<37 weeks’ gestation), low birthweight (LBW—birthweight < 2500 g) and perinatal mortality were compared between groups. Inadequate prenatal care was defined as fewer than 3 visits in any prenatal care facility during pregnancy. Hospitalizations incidence of the offspring up to the age of 18 years with any respiratory, neurological, endocrine and infectious diagnoses morbidities (pre-defined by a set of ICD-9 codes detailed in the on-line Appendix A) were evaluated taking into account the follow-up time. Follow-up was terminated if any of the following occurred: first hospitalization with any of the above mentioned morbidities, hospitalization resulting in death, end of the study period or when the child reached 18 years of age. Multiple pregnancies and fetuses with congenital malformations or chromosomal abnormalities were excluded from the cohort. Perinatal mortality cases (including intra-uterine fetal demise, intra-partum death and neonatal death up to 7 days post-partum) were excluded from the long-term analysis. 

### Statistical Analysis

The analysis was carried out using the SPSS package 23rd ed. (IBM/SPSS, Chicago, IL, USA). Categorical data are shown in counts and rates and the differences were assessed by chi-square for general associations. The student’s *t*-test was used for comparison of continuous variables with normal distribution. For perinatal mortality outcome, a multivariable generalized estimating equation (GEE) analysis was constructed to account for siblings and other confounding factors, including parity, smoking status, obesity, gestational age, ethnicity, maternal diabetes and maternal hypertensive disorders. For the long-term outcomes, Kaplan–Meier survival curves were used to compare cumulative hospitalization incidences over time among the study groups (using log-rank test). Only the first admission with any of the above mentioned morbidities-related condition for a given individual was included in the survival analysis. A cox regression model was used to reveal an independent association between unplanned out-of-hospital delivery and future incidence of respiratory, neurological, endocrine or infectious-related hospitalizations of the offspring while adjusting for possible confounding variables, including maternal age, parity, gestational age, ethnicity, smoking status, obesity, maternal diabetes or hypertensive disorders. In-hospital deliveries were the reference group. All analyses were two-sided and a *p*-value of ≤0.05 was considered statistically significant. 

## 3. Results

During the years of the study, 243,682 deliveries that met the inclusion criteria were registered in SUMC, of which 3580 (1.5%) were unplanned out-of-hospital deliveries.

Table 1 summarizes the maternal characteristics of the study population. Most out-of-hospital births occurred in multiparous women (95.9% vs. 76.1%; *p* < 0.01). About a quarter of these women had inadequate prenatal care (26.2%), compared to only 8.7% of women who delivered in-hospital (*p* < 0.01). Women who delivered out-of-hospital were less likely to be diagnosed with diabetes (pre-gestational or gestational) or hypertensive disorder of pregnancy (chronic, gestational hypertension or pre-eclampsia) as compared with women who delivered in-hospital (2.4% vs. 5.0%, *p* < 0.01 and 1.2% vs. 5.1%, *p* < 0.01; respectively). Obesity and smoking was less common in women who had out-of-hospital delivery (0.4% vs. 1.0%, *p* < 0.01 and 0.3% vs. 1.0%, *p* < 0.01; respectively).

Table 2 shows the short-term perinatal outcomes for both groups. Gestational age at birth and birthweight were lower for those born out-of-hospital (38.8 vs. 39.1, *p* < 0.01 and 3060 g vs. 3207 g, *p* < 0.01; respectively). Preterm birth rates were higher for out-of-hospital births (9.1% vs. 6.8%; *p* < 0.01) and those babies had higher rates of LBW as compared with in-hospital birth (11.6% vs. 6.7%; *p* < 0.01). Perinatal mortality rate was significantly higher for out-of-hospital births as compared with in-hospital births (15/1000 vs. 5/1000; OR = 2.90, *p* < 0.01).

In the GEE model (Table 3), adjusted for siblings and other confounders (listed above), unplanned out-of-hospital delivery was found to be an independent risk factor for perinatal mortality (adjusted OR = 2.59, 95% CI 1.91–3.52, *p* < 0.01).

The long-term morbidities that were evaluated and related hospitalization rates of the offspring are presented in Table 4. The total hospitalization rates involving respiratory, neurological or infectious morbidities were lower in children born out-of-hospital as compared to children born in-hospital. Endocrine-related hospitalization rate was comparable between groups. 

The Kaplan–Meier survival curves (shown in Figure 1) demonstrate a significantly lower cumulative incidence of hospitalizations involving any of the above morbidities of children born out-of-hospital as compared with children born in-hospital (log rank *p* < 0.001 for respiratory, neurological and infections morbidities and log rank *p* = 0.04 for endocrine morbidity). The Cox regression models that were used for the respiratory, neurological and infectious morbidities showed lower rates of future hospitalization of the offspring due to the evaluated morbidities in the out of hospital group. The total hospitalization cases for endocrine morbidity were too small to conduct a multivariable analysis.

## 4. Discussion

This large population-based study reaffirms that unplanned out-of-hospital births carry a significant risk for perinatal mortality, requiring the attention of healthcare providers to identify pregnant women at risk for out-of-hospital delivery and emphasize the possible adverse outcomes to their fetus. Similarly to previous publications [1,4], we found the risk of perinatal mortality to be up to 3 folds higher in out-of-hospital deliveries compared with in-hospital deliveries. While elective home birth as a pre-planned event attended by trained midwives and physicians may have comparable outcomes to planned in-hospital birth [10,11,12,13], the accidental, un-intended out-of-hospital delivery, which is the focus of this study, carries higher risk for mothers and babies.

The risk factors for out-of-hospital in our cohort were in accordance to those previously suggested in the literature and consisted mainly of multiparity and inadequate prenatal care. The majority of patients in our cohort (95.9%) who delivered out of hospital were multiparas, which is obviously explained by the faster nature of repeated deliveries. The shorter interval between the initiation of labor and actual birth in multiparous women sometimes results in unplanned, unattended delivery before reaching the hospital. Another important risk factor in our study was Bedouin ethnicity, that is also associated with inadequate prenatal care, as will be discussed here. To note, women who delivered in-hospital had higher rates of GDM and hypertensive disorders than women who delivered out-of-hospital. Since these morbidities are related to offspring morbidity, we were careful to include them in the Cox regression model and the lower incidence of hospitalizations in the out-of-hospital deliveries remained significant, regardless of these conditions. One can also speculate that the lower rate of these complications in the out-of-hospital deliveries group is related to under-diagnosis of these women who had higher rates of inadequate prenatal care and not due to actual lower hypertensive disorders incidence.

The long-term results of our study should be interpreted with great caution, as they seemingly suggest lower rates of long-term hospitalizations in offspring delivered out of hospital. There could be no plausible medical or biological explanation for reduced morbidity in children born out-of-hospital, unless interpreted considering their demographic or socio-economic status, which is closely related to inadequate prenatal care [14].

Worldwide, there are many social, maternal, financial and structural variables that act as barriers to receive adequate prenatal care. Lack of insurance and low socioeconomic and educational level have been cited in most studies as major determinants of prenatal care [15,16,17,18,19]. Maternal issues such as unintended pregnancy and negative feelings about the pregnancy [20], fear of medical procedures or disclosing the pregnancy to others, depression, multiparity and a belief that prenatal care is unnecessary are also important factors to neglected prenatal care [17]. Structural barriers that were suggested include long wait times, the location and hours of the clinic and the cost of services among others [21,22]. Moreover, demographically, ethnic minorities worldwide are at increased risk for underutilization of prenatal care [23]. It has been postulated before by our group [14] that patients with inadequate prenatal care will later underutilize medical services for their offspring. This assumption might clarify our results of the long-term lower risk for hospitalization in children born out-of-hospital.

This study was conducted in Israel, which is a relatively small country (geographically) with a majority of public hospitals and a healthcare system that is based on a National Health Insurance Law (since 1995), which mandates all citizens of the country to join one of four official non-profit health insurance organizations, enabling availability and accessibility to the vast majority of the population and overcoming many of the above-mentioned barriers to receive adequate prenatal care. Nevertheless, the Negev region in southern Israel, where SUMC is located, is populated with the Bedouin population, which has unique cultural and sociodemographic characteristics. Characterized as a semi-nomad way of life, the Bedouin minority is a Muslim community which suffers from an indigent economic base [24]. The Bedouin tradition attributes great importance to familial and tribal cohesiveness and to fertility [25]. It is a very patriarchal community; most women are poorly educated and do not have access to employment outside the home but like all Israeli citizens, Bedouins are covered by National Health Insurance and receive maternal and child preventive services, including prenatal care by the Ministry of Health local Mother and Child Health Clinics, theoretically overcoming the economic barrier. Bedouin women are usually very religious, the vast majority of them are married, and drugs and alcohol are culturally prohibited, so these causes for underuse of prenatal care are virtually not existent. It is possible that the women in this study, whether Bedouin or Jewish, received prenatal care by private clinics and data were not recorded. However, private prenatal care is expensive, so it is unlikely that marginalized communities would choose to adhere to private practices.

It is assumed that these women do not use pre- and postnatal care resources in the same manner as the general population. The available prenatal services are underused by this population [26], possibly owing to a cultural gap, distrust of the service providers, geographical distance to healthcare services, including available prenatal care services, and patriarchal restriction of female autonomy [25,27,28]. In light of our data that about three-quarters (76%) of children born out-of-hospital were Bedouins and the majority (95.9%) born to multiparous women, it is suggested that these mothers also underutilize healthcare for their offspring, resulting in a lower risk for future hospitalizations due to any of the morbidities evaluated in this study.

A major strength of our study is the fact that it is based on a single tertiary hospital that serves the entire Negev population. Thus, it is based on a large non-selective cohort. SUMC provides maternity and pediatric services. It is assumed that unless people migrate outside the Negev area, they are expected to be hospitalized in SUMC. This allows us to follow children born in our hospital that are later hospitalized by combining the two databases—obstetrical and pediatric. The most important limitation to be addressed regarding the study results is that it is based on hospitalization data, so cases included are probably the more severe ones requiring increased surveillance. Since many pediatric illnesses are usually transient or require only short-term treatment, any morbidity dealt with in an ambulatory setting would not be calculated and evaluated.

As a retrospective study, we could not verify that all out-of-hospital deliveries were accidental in nature and not pre-planned home birth, but in contrast to other countries, post-delivery hospitalization is funded by the state—for women and their offspring, and families actually get paid (child-birth allowance) for in-hospital deliveries. Hence, we believe that most of these women do seek in-hospital care and accidently deliver out-of-hospital.

Another limitation is our inability to extract more sociodemographic data on the mother and offspring that can act as barriers for utilization of medical services and probably shed some more light on our findings. Nevertheless, the lower incidence of hospitalizations of offspring born out-of-hospital involving any of the evaluated morbidities, leads to the logical conclusion that the underutilization of healthcare services during pregnancy (even when given for free under state laws) can suggest future similar pattern of poor compliance to medical care for the offspring. Healthcare givers should be aware to this disparity between populations and consider any encounter with the healthcare system as an opportunity to educate patients and encourage them to attain adequate treatment whenever possible. Israel, not unlike other highly developed countries with socialized healthcare, continues to be challenged by the rural, remote, marginalized among its population. Strategies to identify those at the highest risk should be employed for optimizing parental education and care in this group of patients, given the relatively easy accessibility to subsidized prenatal care.

## Figures and Tables

**Figure 1 jcm-09-00339-f001:**
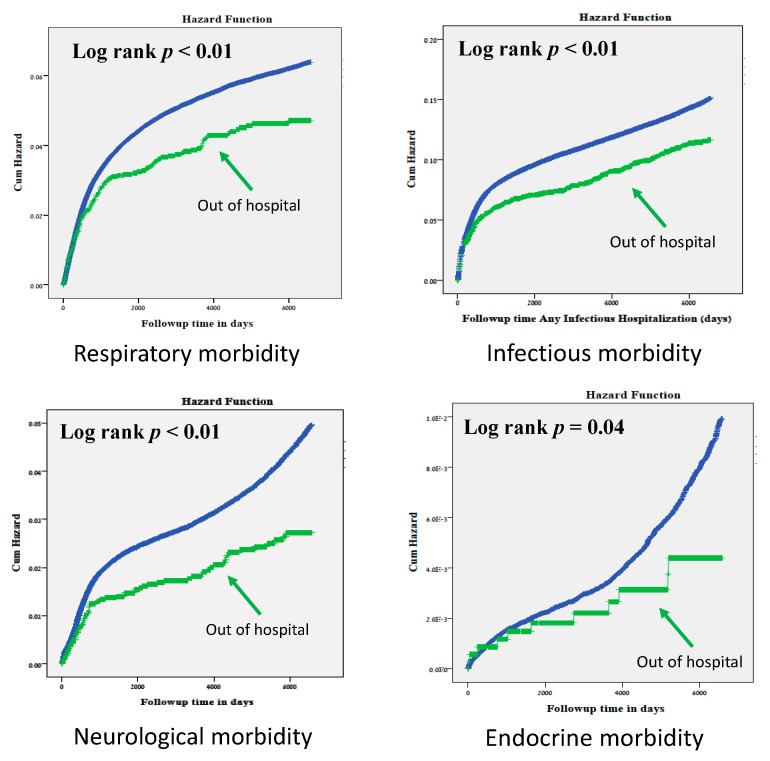
Kaplan–Meier survival curves of the cumulative incidence of hospitalizations for various morbidities among offspring of mothers who had unplanned out-of-hospital delivery compared with in-hospital deliveries.

**Table 1 jcm-09-00339-t001:** Maternal characteristics of mothers with unplanned out-of-hospital delivery compared to in-hospital deliveries.

Maternal Characteristics	Out-of-Hospital Delivery *n* = 3580	In-Hospital Delivery *n* = 240,102	*p* Value
Maternal age (year)	28.4 ± 5.8	28.1 ± 5.8	<0.01
Ethnicity			<0.01
Jewish	860 (24.0%)	114,494 (47.7%)	
Bedouins	2720 (76.0%)	125,608 (52.3%)	
Parit			<0.01
Primiparity	147 (4.1%)	57,358 (23.9%)	
+2	3430 (95.9%)	182,694 (76.1%)	
Diabetes Mellitus ^a^ (*n*)	85 (2.4%)	12,074 (5.0%)	<0.01
Hypertensive Disease ^b^ (*n*)	42 (1.2%)	12,205 (5.1%)	<0.01
Inadequate Prenatal Care ^c^ (*n*)	938 (26.2%)	20,939 (8.7%)	<0.01
Smoking ^d^ (*n*)	12 (0.3%)	2472 (1.0%)	<0.01
Obesity ^e^ (*n*)	13 (0.4%)	2470 (1.0%)	<0.01

^a^ including pre gestational and gestational diabetes. ^b^ including pre gestational, gestational hypertension, and pre-eclampsia. ^c^ fewer than 3 visits to any prenatal care facility during pregnancy. ^d^ by self-report. ^e^ body mass index (BMI) > 30 kg/m^2^.

**Table 2 jcm-09-00339-t002:** Pregnancy outcomes for mothers with unplanned out-of-hospital delivery compared to in-hospital deliveries.

Pregnancy Outcome	Out-of-Hospital Delivery *n* = 3580	In-Hospital Delivery *n* = 240,102	*p* Value
Mean Birthweight (g ± SD)	3060.0 ± 523	3207.9 ± 510	<0.01
Mean Gestational Age at Birth (weeks ± SD)	38.8 ± 2.1	39.1 ± 1.9	<0.01
Preterm Delivery ^a^ (*n*)	326 (9.1%)	16,394 (6.8%)	<0.01
Small for Gestational age ^b^ (*n*)	295 (8.2%)	10,995 (4.6%)	<0.01
Low Birthweight ^c^ (*n*)	417 (11.6%)	15,987 (6.7%)	<0.01
Perinatal Mortality (*n*)	55 (1.5%)	1285 (0.5%)	<0.01

^a^ Preterm < 37 weeks of gestational age. ^b^ Small for gestational age (SGA) < 10th percentile for gestational age. ^c^ Low birth weight (LBW) < 2500 g.

**Table 3 jcm-09-00339-t003:** Multivariable generalized estimating equation (GEE) analysis for perinatal mortality outcome.

Variables	Adjusted OR	95% CI		*p* Value
		Min	Max	
Out of Hospital Delivery	2.59	1.91	3.52	<0.01
Gestational Age (Weeks)	0.66	0.65	0.68	<0.01
Ethnicity	0.59	0.52	0.67	<0.01
Diabetes Mellitus	0.72	0.52	0.99	0.04
Hypertensive Disorders of Pregnancy	1.51	1.24	1.84	<0.01
Smoking	0.96	0.57	1.62	0.89
Obesity	1.79	1.07	2.98	0.02
Parity	1.00	0.86	1.16	0.99

**Table 4 jcm-09-00339-t004:** Hospitalization rates among offspring of mothers with unplanned out-of-hospital delivery compared to in-hospital delivery.

Cause of Hospitalization	Out-of-Hospital Delivery *n* = 3580	In-Hospital Delivery *n* = 240,102	*p* Value	Cox Proportional Hazards Model	
				Adjusted Hazard Ratio (95% CI) ^a^	*p* Value
Respiratory Disease	139 (3.9%)	11,609 (4.9%)	0.01	0.68 (0.58–0.81)	<0.01
Infectious Disease	335 (9.1%)	27,224 (11.1%)	<0.01	0.70 (0.63–0.78)	<0.01
Neurological Disease	74 (2.1%)	7469 (3.1%)	<0.01	0.60 (0.48–0.75)	<0.01
Endocrine Disease	11 (0.3%)	1140 (0.5%)	0.15	NA ^b^	NA

^a^ adjusted for maternal age, parity, smoking, obesity, ethnicity, gestational age, maternal diabetes (pre-gestational and gestational) and hypertensive disorders of pregnancy (pre-gestational, gestational hypertension and pre-eclampsia). ^b^ N too small to calculate Hazard ratio.

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
