# Peer review of "Unplanned Out-of-Hospital Birth—Short and Long-Term Consequences for the Offspring"

_jcm, 2020, doi:10.3390/jcm9020339_

Round 1
Reviewer 1 Report
The authors have addressed my comments.
Reviewer 2 Report
the paper is clear and well written
previous reviews have been taken into consideration
the discusssion clearly states the limits of the results
this paper deserves publication
Reviewer 3 Report
This is an interesting paper, with –be it somewhat hypothesis driven but reasonable – family-based substandard health seeking behavior at delivery and throughout subsequent pediatric care ((substandard health seeking behavior is a family trait’).
The abstract does not sufficiently well reflect the fact that in essence all deliveries (have I understood this correct ?) occur in the hospital ?
How has ‘inadequate care’ been defined ?
How should I consider ‘perinatal mortality’ as – I assume intra-uterine death is not co-collected, can you provide some more information on this ? or does this really covers the definition of perinatal death ?
The discussion also mentions maternal issues and morbidity ? Do you have an data on this ? Along the same line, do you have any ‘outcome’ data in these children ? like school attendance, performance, growth patterns ? immunization/vaccination aspects ?
minor: the figure 2 quality should be improved
Author Response
Please see the attachment

This manuscript is a resubmission of an earlier submission. The following is a list of the peer review reports and author responses from that submission.
Round 1
Reviewer 1 Report
The results of this study will be of interest to a wide readership and advance what is known in this field. The rationale for the study is strong and the methods, including analyses, are appropriate for the most part. The results raise some questions, however, that have not been addressed by the authors in their discussion or conclusions. Some of these are a result of the distinct population from which most of the out-of-hospital women belong.
Queries are below.
Methods/Analyses
1. At what point after delivery were out-of-hospital women brought to the hospital?
2. Is it possible that prenatal care was administered within the ethnic Bedouin community and not recorded?
3. Was smoking recorded?
4. How was gestational age assessed? Is it possible that the out-of-hospital group didn't keep as close track of their periods? How much error might be associated with this variable, particularly in the out-of-hospital group?
5. Student’s t-test is correct (not Student t-test)
6. How was data normality tested?
Results
7. Table 1 - Is age really different between groups?
8. Was severity of hospitalizations different between groups?
Discussion
9. The incidence of maternal gestational diabetes/hypertension were both significantly greater in mothers from the in-hospital group. There is evidence in the literature for these maternal conditions leading to morbidity in offspring. For example, maternal hypertensive disorders strongly affect perinatal mortality. Thus, there is the possibility that higher incidence of these disorders in the in-hospital birth group is causal in the higher hospitalizations in offspring from this group. They remained significant even in the GEE analysis, with hypertensive disorders having a strong adjusted OR. Why is there no discussion of the possibility that the lower hospitalizations in offspring from out-of-hospital births is related to their moms' lower hypertensive disorders?
Reviewer 2 Report
The paper is very well written but the initial premises are incorrect. It's inappropriate to use retrospective population studies with Kaplan-Meier survival curves due to the lack of an RCT. Rather than measuring the survival rate of unhospitalized birth, the study measures the survival rate of birth that is not culturally engaging, lack of prenatal care, the effects of livin in a rural and remote area and failure of this particular hospital to meet the needs of such a large geographical area. Although it was a strength that longitudinal studies could take place with infants as they developed, it would have been much better to have had outlying community health centres (CHCs) and clinics engaging these Bedoiin communities into regular care, perhaps through the use of the World Health Organization obstetrical passport which an be used at multiple CHCs. The use of mHealth has been tremendous, even in the care of Nomadic peoples. I concede that this is an enormous health policy challenge globally - the engagement of marginalized communities. But the KM survival curve isn't the right tool for the research question. And perhaps the authors had another research question in mind - "What are the short-term and long-term survival outcomes for unplanned birth with poorly engaged communities?".
Unfortunately, I don't think it's appropriate to use Kaplan-Meier survival curves as the test statistic for out-of-hospital births even when "unplanned" is controlled for. There are multiple factors for which there has not been a control. Kaplan=Meier is a non-parametric procedure. I think it's laudible that the author is trying to make a case for controlled, planned birth, regardless of the place of birth and several other authors have looked at planned out-of-hospital birth scenarios in which the care providers were well integrated into the healthcare system, full-timer, low-risk without complications or complex social factors such as lack of access to the social determinants of health. The Snowden study cited is a completely different population in which midwifery care is neither integrated nor regulated successfully so that the planned hospital births were not for similar populations that are found in other parts of the US, UK and Canada. Other large populations studies to consider include: Vedam S, Stoll K, Schummers L, Fairbrother N, Klein MC, Thordarson D, Kornelsen J, Dharamsi S, Rogers J, Liston R, Kaczorowski J. The Canadian birth place study: examining maternity care provider attitudes and interprofessional conflict around planned home birth. BMC pregnancy and childbirth. 2014 Dec;14(1):353; Hutton EK, Reitsma AH, Kaufman K. Outcomes associated with planned home and planned hospital births in low‐risk women attended by midwives in Ontario, Canada, 2003–2006: A retrospective cohort study. Birth. 2009 Sep;36(3):180-9. Hutton, E. Outcomes associates with planne dplace of birth among women with low risk pregnancies. Canadian Medical Association Journal; CMAJ 2015:188.5; Vedam S, Declercq ER, Monroe SM, Joseph J, Rubashkin N. The giving voice to mothers study: measuring respectful maternity Care in the United States [18Q]. Obstetrics & Gynecology. 2017 May 1;129(5):177S.Vedam S, Declercq ER, Monroe SM, Joseph J, Rubashkin N. The giving voice to mothers study: measuring respectful maternity Care in the United States [18Q]. Obstetrics & Gynecology. 2017 May 1;129(5):177S.
It would be more appropriate to use Cox regression to actually rule out the effecto all of the demographic variables listed (distance from hospital, number of prenatal visits, parity, pre-existing medical conditions, health literacy, status using Bedouin measures - which might dictate accessibility of transport to hospital nutrition etc.) rather than retrospective sampling or multiple regression as these other researchers have done. The Cox regression was used to look at long-term outcomes but did not factor in the fact that many of the planned hospitalized births included pre-existing medical complications. There was no mention of the in-hospital birth medical profile and the only demographic mentioned was Israeli citizenship. It was helpful and relevant to know that medical care is universal. However, emerging equity research explains that available resources are not always accessible or perceived by marginalized populations.
This is an important and valuable study. I believe that the data could be re-analyzed and the same messages described in the discussion regarding the need to engage Bedouin communities in a way that provides earlier intervention into their healthcare. Could it also be that Israel, not unlike other highly developed countries with socialized healthcare, continue to be challenged by the rural, remote, marginalized among its population. That is a much stronger message that MDPI readers would like to hear.